# Analysis of the Physico-Chemical Properties of Bean Seeds after Three Years of Digestate Use

Milan Koszel [1], Stanisław Parafiniuk [1,*], Sławomir Kocira [1,2], Andrzej Bochniak [3], Artur Przywara [1], Edmund Lorencowicz [1], Pavol Findura [2,4] and Atanas Zdravkov Atanasov [5]

[1] Department of Machine Operation and Production Processes Management, University of Life Sciences in Lublin, 20-612 Lublin, Poland; milan.koszel@up.lublin.pl (M.K.); slawomir.kocira@up.lublin.pl or skocira@fzt.jcu.cz (S.K.); artur.przywara@up.lublin.pl (A.P.); edmund.lorencowicz@up.lublin.pl (E.L.)

[2] Faculty of Agriculture and Technology, University of South Bohemia in České Budějovice, 37333 Nové Hrady, Czech Republic; pavol.findura@uniag.sk

[3] Department of Applied Mathematics and Computer Science, University of Life Sciences in Lublin, 20-612 Lublin, Poland; andrzej.bochniak@up.lublin.pl

[4] Institute of Agricultural Engineering, Transport and Bioenergetics, Slovak University of Agriculture in Nitra, 949 76 Nitra, Slovakia

[5] Department of Agricultural Machinery, University of Ruse Angel Kanchev, 7017 Ruse, Bulgaria; aatanasov@uni-ruse.bg

* Correspondence: stanislaw.parafiniuk@up.lublin.pl

**Abstract:** Taking into consideration its physico-chemical properties, digestate should be used primarily as a fertiliser. The possible ways of using digestate as a fertiliser in agriculture were identified, and digestate collected from an agricultural biogas plant was tested for its macroelement and heavy metal content. The research was conducted on Haplic LUVISOLS soil according FAO classification. The area of the land plots was 75 m$^2$. All measurements were carried out in ten replicates. Seed yield was determined at 2.6 t ha$^{-1}$. The thousand-seed weight was similar in the three growing seasons, and averaged 171.49 g to 184.44 g for the three years under analysis. For the control object, the average thousand-seed weight from the three years of the experiment was 168.56 g. This parameter was significantly influenced by the year of analysis. The highest protein content was obtained in 2022 (an average of 20.3%), which was significantly higher than in 2021 (20.13%) and 2020 (20.12%). The analysis showed an increase in the average value for the three harvest years regarding the fat content of the multiflora bean seeds depending on the post-harvest digestate dose, ranging from 0.47% to 0.61%. In the control object, the average fat content for the three harvest years under analysis was 0.41%. The year under analysis had no significant impact on fat content. A positive correlation was found between the digestate dose and protein, fat, and carbohydrate contents per 100 g of beans. Increasing the dose resulted in statistically significant differences from the lower dose. The obtained results show an increase in macroelement content depending on the digestate dose applied. The average carbohydrate content per 100 g of beans for the three years under analysis ranged from 49.78 g to 54.01 g, while the calcium content per 100 g of beans ranged from 109.23 mg to 124.00 mg. In contrast, the magnesium content in 100 g of bean ranged from 129.91 g to 137.01 mg, the phosphorus content in 100 g of bean from 366.99 mg to 387.00 mg, and the potassium content in 100 g of bean from 1341.20 mg to 1394.06 mg. Statistical analysis revealed statistically significant differences except for potassium, where no differences were found for the two highest doses. In addition, no differences were found in the average phosphorus and potassium content between the years under analysis. The study showed an increase in yield depending on the amount of digestate applied. The highest dose used in the experiment provided the most nitrogen and macronutrients, with a positive effect on yield velocity, protein and fat content, micronutrients, and macronutrients in beans.

**Keywords:** digestate; fertilisation; bean *Phaseolus coccineus* seeds; protein content; microelements; macroelements

## 1. Introduction

Next to medicine, food occupies the highest position in the bioeconomic value pyramid. Its production is the most vital part of the economic process. In the context of the constantly growing population, it appears indisputable that land resources should be used first and foremost to feed people [1]. Mineral fertilizers are man-made substances. Unfortunately, the use of fertilizers can have harmful effects on the environment. Excessive and inappropriate fertilization is the cause of ground and surface water pollution, resulting in eutrophication, changes in the soil solution (acidification, alkalinization) or salinization of soils, and the accumulation of harmful substances in plants. Fertilizers applied in excess and not used by plants remain in the soil and are washed out with rainwater into water bodies. On the other hand, digestate makes it possible to significantly reduce mineral fertilization inputs, contributing to improving soil quality, which in turn results in higher yields. It is worth remembering that poferment, unlike traditional natural fertilizers (slurry and manure), can be applied to fields all year round except during periods of snow cover and ground frost. This means that by using digestate it is possible to avoid the costs and restrictions associated with compliance with the current nitrate directive. According to this directive, farmers can only apply manure to fields within a certain time frame. This restriction does not apply to digestate, which can be applied to agricultural land for most of the year, that is, when the soil is not frozen to a depth of 30 cm, covered with snow cap, or flooded. In addition, according to the nitrate regulation, less acreage is required for the management of digestate than for manure.

Regulations regarding the production and use of all sorts of fertilisers are becoming increasingly stringent. The use of digestate as a fertiliser can contribute to securing adequate yields while reducing negative environmental impacts. Moreover, biogas production based on waste and byproducts feeds into sustainable development and the circular economy [2].

For fermentation processes, agricultural biogas plants only use byproducts directly originating from agricultural production, such as manure and straw or biowaste from the food industry. In recent years, energy crops, cereals, and maize silage have become major feedstocks for biogas plants [3].

Anaerobic digestion involves the degradation of organic material in the absence of oxygen to produce biogas (a mixture of methane, carbon dioxide and trace amounts of other gases) and a stabilised residual digestate product. This process is growing in popularity, as it allows renewable energy to be produced from a wide variety of municipal, agricultural, and industrial organic waste while enabling nutrient recovery through application of digestate to soils [4].

Legally, biogas digestate from agricultural biogas plants is treated as potentially hazardous waste of the sewage sludge type, which greatly complicates the possibilities around its management. As waste, the digestate can be disposed of; however, recovery is generally recommended. As a waste, digestate is recovered using the R10 process, i.e., surface treatment of waste to benefit agriculture.

Sludge, referred to as biogas digestate or simply digestate, is a secondary product of anaerobic digestion. It contains huge amounts of organic compounds of both plant and microbial origin along with numerous minerals. The key feature of digestate is its low concentration of dry matter, ranging from a few to several percent [3,5].

Digestate from biogas plants is a valuable source of nutrients for crop production [6,7]. Due to its chemical composition and physical properties, the solid digestate phase can exert a positive impact on both biomass yield and soil structure [8,9]. Velechovský et al. [10] described digestate as an organic fertiliser containing mineral nutrients along with organic matter. The use of digestate as a fertiliser contributes to the recycling of organic matter and minerals, and increases the profitability of crop production by reducing fertilisation costs [11,12]. Sienkiewicz et al. [12] considered the spreading of digestate across the field surface to be the simplest and cheapest way to use this post-production waste.

There are two bean species cultivated in the Polish climate: *Phaseolus vulgaris* L. (the common bean) and *Phaseolus multiflorus* Wild (the multiflora bean). Among the common

bean varieties, dwarf beans with a rigid stem length of about 40 cm and pole beans with a limp stem length of up to 3 m can be distinguished. There also exists an intermediate form with a stem length of 60–120 cm and without the ability to wind around the pole.

Beans are rich in protein, potassium, phosphorus, and iron. Phytic compounds lower cholesterol levels, have anti-cancer effects, and help prevent colon cancer.

Research on the effects of the digestate dose in bean upregulation has been undertaken due to the fact that it is one of the most widely grown legumes. It is valued for its high content of protein and B vitamins, particularly by people who limit their meat intake or are on a vegan diet. Beans in the form of green pods are a valuable raw material for canning and freezing, and are a raw material for vegetable processing. The research presented here makes it possible to limit the use of mineral fertilisers, thereby reducing their harmful effect on the environment. Determining the impact of the use of digestate on fertilizer quality is currently an important research problem for ecological and economic reasons. Fertilisers applied in liquid form (such as digestate) enter the soil solution more quickly and are better absorbed by plants.

This study aimed to determine the ideal digestate dosage and to analyse the response of the "Kontra" variety of multiflora bean to different dosages. The hypothesis of this study was that the volume of the digestate dose would affect both the yield and seed quality.

## 2. Materials and Methods

This study focused on field-grown plants in the 2020, 2021, and 2022 growing seasons along with seeds obtained from these plants. Soil type according FAO classification was Haplic LUVISOLS. Soil samples were evaluated. This included determining their pH and macroelement content. In addition, samples of the digestate applied during the cultivation process were analysed to determine their heavy metal content. The tests and analyses were performed in a laboratory in accordance with PN-EN ISO/IEC 17025:2005 [13]. Testing was performed in accordance with the PN-EN 13650 and PN-EN 13654-1 standards [14]. All measurements were carried out in ten replicates.

Meteorological data were obtained from the Institute of Meteorology and Water Management of the National Research Institute.

Seeds of the *Phaseolus coccineus* bean ("Kontra" variety) obtained from PlantiCo were used in the experiments. Soil samples were collected from the arable land of the Experimental Farm (University of Life Sciences in Lublin) in Czesławice (Lublin Province, 51°18′23″ N, 22°16′2″ E). Egner Riehm soil sticks were used to collect the samples from a soil layer of 0 ÷ 20 cm from twelve experimental plots, after which they were mixed to obtain representative samples.

Sowing was carried out on medium soil with sand dust grain size. The experimental land plots covered an area of 75 m$^2$, and the seeds used in the experiments were multiflora bean ("Kontra" variety) seeds obtained from PlantiCo. The plants used in the experiment are classified as a flagellate variety that is very fertile and moderately late-cropping. The plants are of a moderate size and compact type, and display high resistance to legume diseases.

Pre-sowing treatment with Polifoska 6 mineral fertiliser (6% nitrogen (N) content in ammonium form, 20% phosphorous (P$_2$O$_5$) content, and 30% potassium (K$_2$O) content) was applied at the following doses: in 2020, 250 kg ha$^{-1}$; in 2021, 250 kg ha$^{-1}$; and in 2022, 250 kg ha$^{-1}$.

Crop protection treatments were performed with a Pilmet 312 LM suspended field sprayer. The application dose was 270 L ha$^{-1}$, with a working speed of 5.4 km h$^{-1}$ and a working pressure of 3.0 bar. The height of the beam above the sprayed area was 50 cm.

The multiflora bean seeds were sown on a field after winter wheat. The field was then disked and harrowed. These procedures were followed by ploughing at a depth of 20 cm. The field was levelled off with a cultivating unit consisting of a light-tine cultivator and a string roller. The seeds were sown at a depth of 4 cm and at every 10 cm in a row, with a row spacing of 40 cm.

The field tests covered four experimental variants, with the digestate spread using a slurry spreader with hoses arranged across the soil surface in the following manner:

- Variant I—control object; the seeds were sown, but no digestate was applied.
- Variant II—total digestate dose of 25,000 L ha$^{-1}$ (dose 1: 12,500 L ha$^{-1}$; dose 2: 12,500 L ha$^{-1}$),
- Variant III—total digestate dose of 37,500 L ha$^{-1}$ (dose 1: 18,750 L ha$^{-1}$; dose 2: 18,750 L ha$^{-1}$),
- Variant IV—total digestate dose of 50,000 L ha$^{-1}$ (dose 1: 25,000 L ha$^{-1}$; dose 2: 25,000 L ha$^{-1}$).

The above doses were determined based on soil sample analyses and studies performed by other authors [15,16].

The digestate was applied in the spring in two doses:

- Dose 1 at BBCH 10 (cotyledons completely unfolded;,
- Dose 2 at BBCH 13 (third true leaf unfolded).

The digestate was collected from the Piaski biogas plant (Lublin Province, 51°8′16″ N, 22°53′39″ E). The following raw materials were used for digestate production: maize silage (70%), sugar beet pulp (15%), fruit pomace (5%), milk waste (5%), and manure (5%). The digestate was applied in liquid form using spill hoses.

At the end of September, when the leaves had yellowed and fallen and the pods and seeds had reached maturity, the harvesting process was performed manually. The plants were uprooted and left to dry. After cleaning, the seeds were stored in jute bags for two weeks in laboratory rooms at a temperature of 20 °C. They were stirred every day to ensure that the moisture content was distributed evenly, then analysed.

The post-harvest seed quality tests involved determining the seed moisture, thousand-seed weight (TSW), protein content, fat content, carbohydrate content, and calcium, magnesium, phosphorus, and potassium contents. The above tests were performed at the ISO 17025 [13] accredited Central Agroecological Laboratory of the University of Life Sciences in Lublin.

The moisture content of the seeds tested was determined at 9.97% in 2020, 11.04% in 2021, and 10.02% in 2022. The protein content was established using the titration method (Kjeldahl). When the mineralisation process is completed, the nitrogen contained in the sample takes the form of acidic ammonium sulphate. Subsequently, the protein content is determined using an appropriate conversion factor. All measurements were carried out in ten replicates.

The fat content was determined using the Soxhlet extraction method. Testing was performed in accordance with the PN-EN ISO 11085:2015-10 standard [17]. All measurements were carried out in ten replicates.

The phosphorus content was determined using the spectrophotometric method. Testing was performed in accordance with the PN-EN ISO 3946:2000 standard [18]. All measurements were carried out in ten replicates.

The carbohydrate content was determined using high-pressure liquid chromatography (HPLC). The identification process was carried out after separation on a suitable chromatographic column, and the amount of individual carbohydrates in the samples was calculated based on the retention times and areas of the separated peaks. Testing was performed in accordance with the PN-EN ISO 16181:2018-09 standard [19]. All measurements were carried out in ten replicates.

The calcium, magnesium, and potassium contents were determined using atomic absorption spectrophotometry (ASA). The measured absorption was proportional to the concentration of the element concerned. Testing was performed in accordance with the PN-EN ISO 1134:1999 standard [20]. All measurements were carried out in ten replicates.

Pod length and width, pod count per plant, and seed count per plant were all measured after harvesting. All measurements were carried out in ten replicates.

Statistical analysis was performed using the generalised linear model (GLM). The analysed indicators (variables) included pod length and width, pod count and seed count per plant, yield, thousand-seed weight, and content of protein, fat, carbohydrate, calcium, magnesium, phosphorus, and potassium per 100 g of beans. The following factors were used as explanatory factors: the digestate rate (D) as a fixed factor; the growing year (Y) as a random factor; and the interaction between the digestate rate and the growth year (DxY) as a random factor. When significant differences were found in the obtained results, Tukey's post hoc Honest Significant Difference (HSD) tests were used to determine which groups (doses or years) exhibited statistically significant differences.

Statistical analysis was performed using Tibco Statistica v. 14.0 with a significance level of $\alpha = 0.05$.

### 3. Results

The meteorological conditions prevailing in the cultivation period of the multiflora beans in Lublin–Radawiec (51°12′26″ N, 22°23′9″ E) were analysed.

Figure 1 presents the weather conditions, including air temperature and precipitation, during the multiflora bean cultivation period.

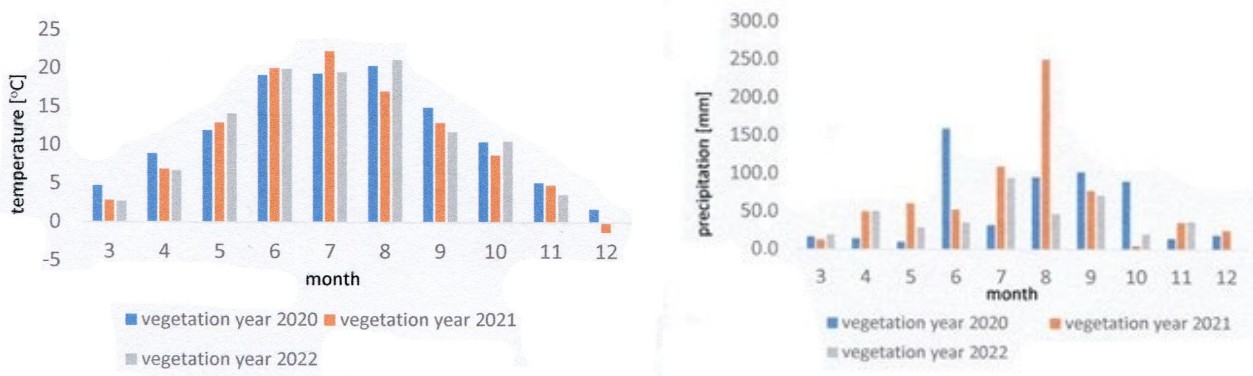

**Figure 1.** Weather conditions in 2020–2022.

All the three growing seasons under analysis were similar in terms of average monthly air temperature. In terms of precipitation, the 2020 and 2021 growing seasons were similar, while the 2022 growing season was clearly drier.

Prior to application, the digestate was tested for macronutrients and heavy metals (Table 1). The pH of the digestate applied under mung bean cultivation was 8.65, which is similar to that of cattle slurry (7.9).

The heavy metal content was found to be too low to be detected by the measuring instruments. The digestate contained substantial amounts of macroelements, and as such was considered viable as a fertiliser. Based on the Regulation of the European Parliament and the EU council of 5 June 2019, the content of heavy metals in organic fertilizer must not exceed the following: cadmium—1.5 mg kg$^{-1}$ dry weight; lead—120 mg kg$^{-1}$ dry weight; nickel—50 mg kg$^{-1}$ dry weight; chromium—2 mg kg$^{-1}$ dry weight. In organic–mineral fertilizer, a higher cadmium content of 3 mg kg$^{-1}$ dry weight is allowed. The same cadmium content is allowed in inorganic fertilizer. A higher content of nickel, 100 mg kg$^{-1}$ dry weight, is allowed in this fertilizer.

The analysis of the soil for acidity and macroelement content was carried out before sowing and after harvesting the multiflora bean seeds (Tables 2 and 3).

**Table 1.** Comparison of selected macroelements and heavy metals in the digestate.

| Analysed Feature | Digestate for Bean Seed Cultivation | | | | Standard Deviation |
|---|---|---|---|---|---|
| | **2020** | **2021** | **2022** | **Average** $\bar{x}$ | |
| Nitrogen (g L$^{-1}$) | 4.22 | 4.15 | 4.24 | 4.20 | 0.05 |
| Phosphorus (g L$^{-1}$) | 0.16 | 0.15 | 0.19 | 0.17 | 0.02 |
| Potassium (g L$^{-1}$) | 5.38 | 5.25 | 5.40 | 5.34 | 0.08 |
| Calcium (g L$^{-1}$) | 0.34 | 0.32 | 0.35 | 0.34 | 0.02 |
| Magnesium (g L$^{-1}$) | 0.09 | 0.07 | 0.10 | 0.09 | 0.02 |
| Cadmium (mg L$^{-1}$) | <0.43 | <0.43 | <0.43 | <0.43 | 0 |
| Lead (mg L$^{-1}$) | <0.43 | <0.43 | <0.43 | <0.43 | 0 |
| Nickel (mg L$^{-1}$) | <0.43 | <0.43 | <0.43 | <0.43 | 0 |
| Chromium (mg L$^{-1}$) | <0.43 | <0.43 | <0.43 | <0.43 | 0 |
| Copper (mg L$^{-1}$) | 0.48 | 0.44 | 0.51 | 0.48 | 0.04 |
| Zinc (mg L$^{-1}$) | 1.99 | 2.03 | 2.07 | 2.03 | 0.04 |
| Manganese (mg L$^{-1}$) | 2.17 | 2.21 | 2.26 | 2.21 | 0.05 |
| Iron (mg L$^{-1}$) | 77.64 | 79.23 | 76.34 | 77.74 | 1.45 |

**Table 2.** pH and macroelement content tests in the Haplic LUVISOLS soil type prior to multiflora bean cultivation.

| Year | Acidity | Phosphorus | Potassium | Magnesium |
|---|---|---|---|---|
| | **(pH)** | **(mg 100 g$^{-1}$ Soil)** | **(mg 100 g$^{-1}$ Soil)** | **(mg 100 g$^{-1}$ Soil)** |
| 2020 | 7.10 | 47.59 | 48.22 | 11.03 |
| 2021 | 6.95 | 47.56 | 48.69 | 10.57 |
| 2022 | 7.05 | 47.73 | 48.25 | 10.72 |
| Average $\bar{x}$ | 7.03 | 47.63 | 48.39 | 10.77 |

**Table 3.** pH and macroelement content tests in the Haplic LUVISOLS soil type after multiflora bean cultivation.

| Year | Acidity | Phosphorus | Potassium | Magnesium |
|---|---|---|---|---|
| | **(pH)** | **(mg 100 g$^{-1}$ Soil)** | **(mg 100 g$^{-1}$ Soil)** | **(mg 100 g$^{-1}$ Soil)** |
| 2020 | 6.95 | 24.02 | 34.69 | 9.41 |
| 2021 | 6.80 | 24.29 | 33.15 | 8.34 |
| 2022 | 6.70 | 24.36 | 34.47 | 9.50 |
| Average $\bar{x}$ | 6.82 | 24.22 | 34.10 | 9.08 |

Soil acidity (pH) in the control field after multiflora bean harvesting reached 6.50 in 2020, 6.50 in 2021, and 6.55 in 2022, while the macroelement content was as follows:

- Phosphorous—20.2 mg 100 g$^{-1}$ soil in 2020, 16.5 mg 100 g$^{-1}$ soil in 2021, and 18.4 mg 100 g$^{-1}$ soil in 2022;
- Potassium—19.2 mg 100 g$^{-1}$ soil in 2020, 21.0 mg 100 g$^{-1}$ soil in 2021, and 20.5 mg 100 g$^{-1}$ soil (in 2022);
- Magnesium—7.6 mg 100 g$^{-1}$ soil in 2020, 7.3 mg 100 g$^{-1}$ soil in 2021, and 7.5 mg 100 g$^{-1}$ soil in 2022.

The average soil acidity (pH) before sowing determined for the three years of growth under analysis reached 7.03 (neutral pH). After harvesting, the soil pH reached 6.82 (neutral pH). The average phosphorus content in the soil after harvesting decreased by 23.4 p.p. (percentage points) and the average potassium content decreased by 14.3 p.p., while the average magnesium content increased by 1.7 p.p. The percentage changes for the examined macroelements were as follows: the phosphorus content decreased by 49.2%, the potassium content by 29.5%, and the magnesium content by 15.7%.

### 3.1. Influence of the Fertilisation Rate on the Physical Parameters of Multiflora Bean Plants

The average plant density (from ten measurements) per 1 m$^2$ was 51 in 2020, 48 in 2021, and 53 in 2022.

Table 4 shows the results obtained from the adjusted GLM models. For the majority of the parameters under analysis, a very high fit index (R$^2$$_{adj}$) exceeding 0.95 was obtained. Lower values were obtained only for pod width and pod count per plant. For all the parameters, statistically significant differences were found in the average values based on the applied digestate dose (in all cases, *p*-value < 0.001). Significant differences were found in the average values of pod width, pod count per plant, yield, thousand-seed weight, protein content, carbohydrate content, and magnesium content in the successive years of the study. Similarly, as regards certain parameters, different effects of the applied digestate rates were found in successive years (i.e., a significant DxY interaction).

**Table 4.** Results of GLM analysis of the trial with beans in Haplic LUVISOLS soil type, showing the value of the test statistic (F), degrees of freedom (d.f.), *p*-value (*p*), and model fit coefficient (R$^2$$_{adj}$).

| Parameter | Units | Dose (D) d.f. = 3 | Vegetation Year (Y) d.f. = 2 | DxY d.f. = 6 | R$^2$$_{adj}$ |
|---|---|---|---|---|---|
| Average pod length | (cm) | F = 2010.9, $p < 0.001$ * | F = 4.1, $p = 0.076$ | F = 20.0, $p < 0.001$ * | 0.999 |
| Average pod width | (cm) | F = 49.2, $p < 0.001$ * | F = 8.2, $p = 0.019$ * | F = 0.95, $p = 0.466$ | 0.557 |
| Pod count per plant | (pcs) | F = 8821.0, $p < 0.001$ * | F = 1083.0, $p < 0.001$ * | F = 0.014, $p = 0.999$ | 0.770 |
| Pod count per plant | (pcs) | F = 466.0, $p < 0.001$ * | F = 0.0, $p = 0.986$ | F = 4.0, $p = 0.002$ * | 0.978 |
| Yield | (t ha$^{-1}$) | F = 3404.3, $p < 0.001$ * | F = 7.4, $p = 0.024$ * | F = 1.2, $p = 0.310$ | 0.990 |
| Thousand seed weight | (g) | F = 190.9, $p < 0.001$ * | F = 14.2, $p = 0.005$ * | F = 7.38, $p < 0.001$ * | 0.974 |
| Protein content | (%) | F = 1045.4, $p < 0.001$ * | F = 10.0, $p < 0.001$ * | F = 10.6, $p < 0.001$ * | 0.996 |
| Fat content | (%) | F = 163.8, $p < 0.001$ * | F = 2.24, $p = 0.188$ | F = 6.55, $p < 0.001$ * | 0.965 |
| Carbohydrate content in 100 g of bean | (g) | F = 115,108.0, $p < 0.001$ * | F = 7.0, $p = 0.026$ * | F = 0.0, $p = 1.0$ | 0.959 |
| Calcium content in 100 g of bean | (mg) | F = 10,369.0, $p < 0.001$ * | F = 1.0, $p = 0.323$ | F = 32.0, $p < 0.001$ * | 0.999 |
| Magnesium content in 100 g of bean | (mg) | F = 150,135.0, $p < 0.001$ * | F = 6.0, $p = 0.039$ * | F = 1.0, $p = 0.330$ | 0.999 |
| Phosphorus content in 100 g of bean | (mg) | F = 19,317.3, $p < 0.001$ * | F = 0.4, $p = 0.684$ | F = 2.6, $p = 0.022$ * | 0.999 |
| Potassium content in 100 g of bean | (mg) | F = 194,573.9, $p < 0.001$ * | F = 0.4, $p = 0.727$ | F = 1.6, $p = 0.163$ | 0.999 |

* Significant statistical differences.

Table 5 outlines changes in pod length and width in the individual experimental variants.

**Table 5.** Changes in pod length and width in the individual experimental variants of the trial with beans in Haplic LUVISOLS soil type.

| Variant | Average Pod Length (cm) | | | | Average Pod Width (cm) | | | |
|---|---|---|---|---|---|---|---|---|
| | 2020 | 2021 | 2022 | Average $\bar{x}$ | 2020 | 2021 | 2022 | Average $\bar{x}$ |
| I | 8.66 | 8.80 | 8.77 | 8.74 | 0.63 | 0.62 | 0.64 | 0.63 |
| II | 9.68 | 9.70 | 9.78 | 9.72 | 0.65 | 0.66 | 0.67 | 0.66 |
| III | 10.41 | 10.40 | 10.49 | 10.43 | 0.66 | 0.67 | 0.69 | 0.67 |
| IV | 11.00 | 11.01 | 11.01 | 11.01 | 0.69 | 0.70 | 0.70 | 0.70 |

The highest average pod length was observed in variant IV plants in all growing periods. The average pod length in the first vegetation season (2020) ranged from 9.68 cm to 11.00 cm, and was 8.66 cm for the control object. In the second vegetation season (2021), the average pod length ranged from 9.70 cm to 11.01 cm, and was 8.80 cm for the control object. In contrast, in the 3third vegetation season (2022) these values ranged from 9.78 cm to 11.01 cm, with 8.77 cm for the control object.

In Figure 2A, slight variability of pod length can be observed for the years under analysis, with no marked differences between individual years ($p = 0.076$). The lowest average pod length was observed in 2020 (9.94 cm) and the highest in 2022 (10.01 cm).

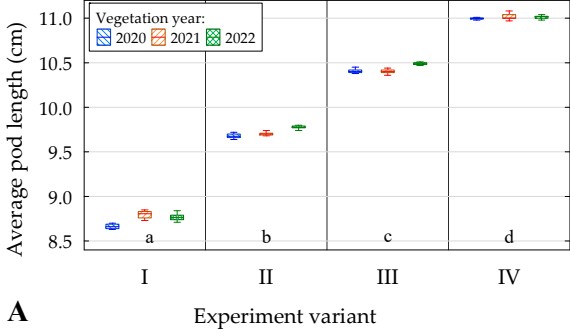 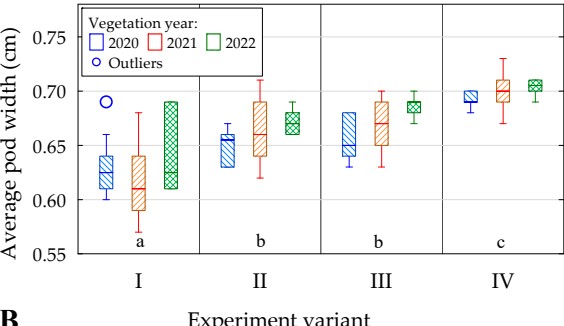

**Figure 2.** Boxplots for the size of bean pods in the trial with beans in Haplic LUVISOLS soil type, showing (**A**) length and (**B**) width. [abcd]—mean values in the groups not containing the same letter differ significantly at $\alpha = 0.05$.

The highest average pod width was observed in variant III plants in all growing periods. The average pod width in the first vegetation season (2020) ranged from 0.65 cm to 0.69 cm, and was for the control object. In the second growing season (2021), the average pod width ranged from 0.66 cm to 0.70 cm, and was 0.62 cm for the control object. In contrast, in the third growing season (2022), the values ranged from 0.67 cm to 0.70 cm, with 0.64 cm for the control object.

There were no significant differences in pod width for digestate doses of 25,000 L ha$^{-1}$ and 37,500 L ha$^{-1}$ ($p = 0.38$). Pod width varied between the successive years of the experiment, with the widest pods being recorded in 2022. The average value of 0.676 cm recorded in that year was significantly higher than in 2020 (0.657 cm, $p < 0001$) or 2021 (0.663 cm, $p = 0.039$) (Figure 2B).

The changes in average pod count and average seed count of the multiflora beans are presented in Table 6.

**Table 6.** Changes in the average pod count and seed count in individual experimental variants of the trial with beans in Haplic LUVISOLS soil type.

| Variant | Average Pod Count per Plant (pcs) | | | | Average Seed Count per Plant (pcs) | | | |
|---|---|---|---|---|---|---|---|---|
| | **2020** | **2021** | **2022** | **Average $\bar{x}$** | **2020** | **2021** | **2022** | **Average $\bar{x}$** |
| I | 7.0 | 7.0 | 8.0 | 7.3 | 25.0 | 25.0 | 26.0 | 25.3 |
| II | 8.0 | 8.0 | 9.0 | 8.3 | 31.0 | 31.0 | 31.0 | 31.0 |
| III | 10.0 | 10.0 | 10.9 | 10.3 | 40.0 | 39.2 | 40.0 | 39.7 |
| IV | 11.0 | 11.0 | 11.9 | 11.3 | 43.0 | 44.0 | 42.3 | 43.1 |

The average pod count per plant in the first vegetation season was in the range of $8.0 \div 11.0$ pcs, and was 7.0 for the control object. In the second vegetation season, the average pod count per plant ranged from 8.0 to 11.0, and was 7.0 for the control object. Finally, in the third vegetation season, the average pod count per plant ranged from 9.0 to 11.0, and was 8.0 for the control object.

The average pod count per plant (Figure 3A) in variant IV reached 7.3, while it was 8.3 in variant I (a rise of 13.6%), 10.3 in variant II (a rise of 40.5% in relation to the control object), and 11.3 in variant III (a rise of 54.1%). In the entire analysis period, the highest average value was recorded in 2022 (9.5), which was significantly higher than in the preceding two years (an average of 9.0, $p < 0.001$ in both cases).

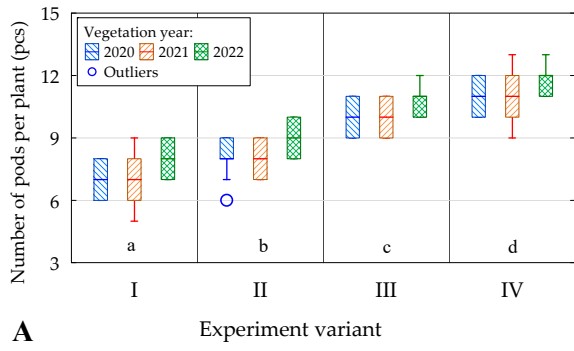
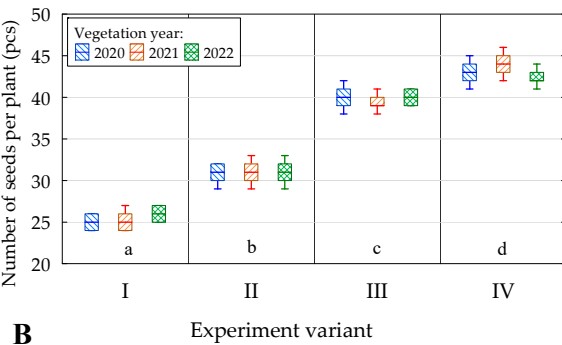

**Figure 3.** Boxplots for yield size indicators of the trial with beans in Haplic LUVISOLS soil type. (**A**) number of pods per plant and (**B**) number of seeds per plant. [abcd]—mean in groups not containing the same letter differ significantly at α = 0.05.

Similar trends to those observed for the average pod count per plant were obtained for the average seed count per plant (Figure 3B). The lowest average seed count per plant was recorded for experimental variant I (an average of 25.3 pods). There was an increase of 22.5% for experimental variant II (an average of 31 pods), 56.8% for variant III (an average of 39.7 pods), and 70.1% for variant IV (an average of 43.1 pods). The year of analysis had a significant impact on the seed count.

### 3.2. Digestate Dose vs. Multiflora Bean Yield

Table 7 outlines changes in yield and thousand-seed weight in the individual experimental variants.

**Table 7.** Changes in yield and thousand-seed weight in individual experimental variants of the trial with beans in Haplic LUVISOLS soil type.

| Variant | Yield (t ha$^{-1}$) | | | | Thousand-Seed Weight (g) | | | |
|---|---|---|---|---|---|---|---|---|
| | **2020** | **2021** | **2022** | **Average $\bar{x}$** | **2020** | **2021** | **2022** | **Average $\bar{x}$** |
| I | 1.80 | 1.80 | 1.82 | 1.81 | 167.71 | 166.82 | 171.15 | 168.56 |
| II | 2.06 | 2.05 | 2.10 | 2.07 | 169.56 | 172.06 | 172.85 | 171.49 |
| III | 2.33 | 2.34 | 2.36 | 2.34 | 177.32 | 178.96 | 179.84 | 178.71 |
| IV | 2.61 | 2.60 | 2.62 | 2.61 | 183.33 | 184.00 | 186.00 | 184.44 |

A discrepancy was found in the yield obtained in the first growing season (2020), ranging from 2.06 t ha$^{-1}$ to 2.61 t ha$^{-1}$. In the second vegetation season (2021), the figures were similar, from 2.05 t ha$^{-1}$ to 2.60 t ha$^{-1}$, while in the third vegetation season (2022), between 2.10 t ha$^{-1}$ and 2.62 t ha$^{-1}$ were harvested depending on the applied digestate dose. The control plot yielded 1.80 t ha$^{-1}$ in the first vegetation season, 1.80 t ha$^{-1}$ in the second growing season, and 1.81 t ha$^{-1}$ in the third growing season.

The average yield in 2022 (2.22 t/ha) was significantly higher than in 2020 (2.2 t ha$^{-1}$) or 2021 (2.197 t ha$^{-1}$) (Figure 4A).

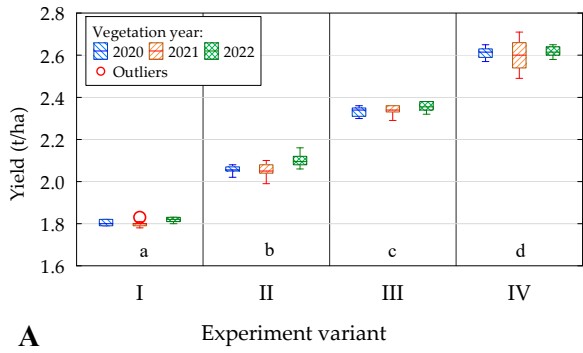
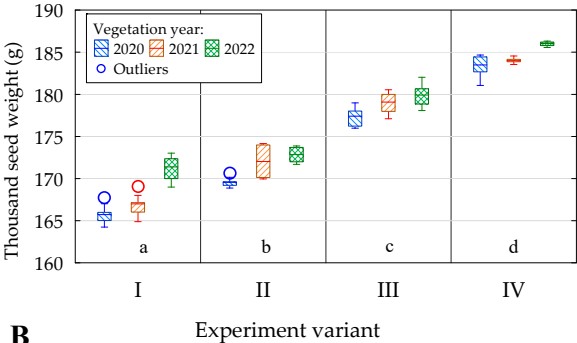

**Figure 4.** Boxplots for yield size indicators of the trial with beans in Haplic LUVISOLS soil type. (**A**) yield and (**B**) thousand-seed weight. abcd—mean in groups not containing the same letter differ significantly at $\alpha = 0.05$.

The thousand seed weight was similar in the three growing seasons, and averaged 171.49 g to 184.44 g over the three years under analysis. For the control object, the average thousand-seed weight from the three years of the experiment was 168.56 g. This parameter was significantly influenced by the year of analysis (Figure 4B).

### 3.3. Impact of Digestate Dose on the Quality of Seeds after Harvesting Multiflora Beans

For the control object, the moisture content of the multiflora bean seeds harvested in 2020 was 11.44%, while in 2021 it was 11.93% and in 2022 it was 11.34%. In the first growing season, the moisture content ranged from 9.86% in experimental variant III to 10.05% in variant II. In the second vegetation season the moisture levels were more even, ranging from 10.85% in experimental variant IV to 11.15% in variant II. The seeds harvested after the third vegetation season had a moisture content ranging from 9.90% in experimental variant IV to 10.13% in variant III.

Table 8 outlines the analysis results regarding the fat and protein contents in multiflora bean seeds in the individual experimental variants.

**Table 8.** Changes in fat and protein contents of multiflora bean seeds in individual experimental variants on Haplic LUVISOLS soil type.

| Variant | Protein Content (%) | | | | Fat Content (%) | | | |
|---------|------|------|------|-----------|------|------|------|-----------|
|         | 2020 | 2021 | 2022 | Average $\bar{x}$ | 2020 | 2021 | 2022 | Average $\bar{x}$ |
| I       | 18.85 | 18.87 | 18.89 | 18.87 | 0.40 | 0.41 | 0.42 | 0.41 |
| II      | 19.42 | 19.41 | 19.70 | 19.51 | 0.47 | 0.48 | 0.47 | 0.47 |
| III     | 20.83 | 20.81 | 21.02 | 20.89 | 0.53 | 0.52 | 0.54 | 0.53 |
| IV      | 21.38 | 21.42 | 21.58 | 21.46 | 0.60 | 0.59 | 0.63 | 0.61 |

The average protein content of the multiflora bean seeds harvested from the control plot was 18.87% for the three years under analysis. In the other experimental variants, the average protein content increased along with the digestate rate, ranging from 19.51% to 21.46%. The highest protein content (Figure 5A) was obtained in 2022 (an average of 20.3%), which was significantly higher than in 2021 (20.13%) or 2020 (20.12%).

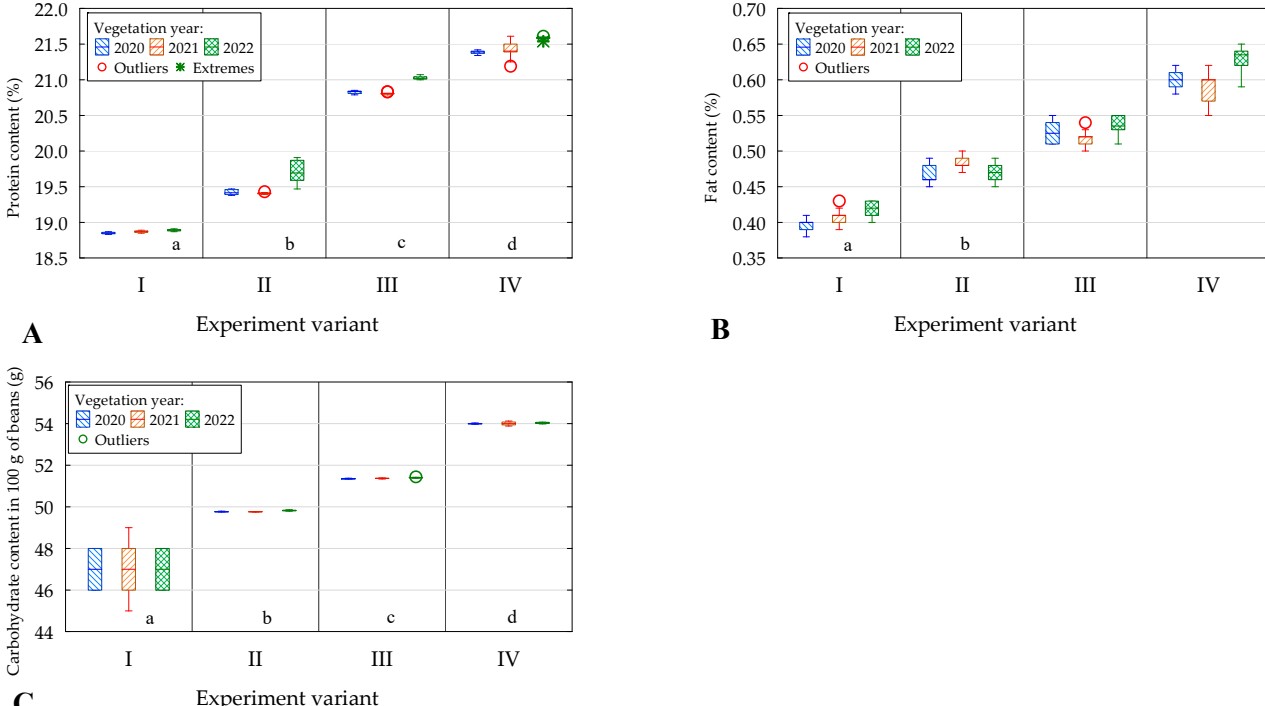

**Figure 5.** Boxplots for contents of (**A**) protein, (**B**) fat, and (**C**) carbohydrates in 100 g of beans grown on Haplic LUVISOLS soil type. [abcd]—mean in groups not containing the same letter differ significantly at $\alpha = 0.05$.

The analysis showed an increase in the average value for the three harvest years regarding the fat content of the multiflora bean seeds. This was dependent on the post-harvest digestate dose, which ranged from 0.47% to 0.61%. In the control object, the average fat content for the three harvest years under analysis was 0.41%. The year of analysis had no significant impact on the fat content (Figure 5B). A positive correlation was found between the digestate dose and the protein, fat, and carbohydrate contents per 100 g of beans (Figure 5A–C). Subsequent increases in the digestate dose resulted in statistically significant differences relative to the lower dose.

Table 9 presents the results regarding changes in the content of macroelements in the multiflora seeds harvested in 2020, 2021, and 2022. Based on these results, the macroelement content in the seeds was found to remain at similar levels throughout the analysis period.

The obtained results show an increase in macroelement content depending on the applied digestate dose (Figure 6A–D). The average carbohydrate content per 100 g of beans (Figure 5C) for the three years under analysis ranged from 49.78 g to 54.01 g, while the calcium content per 100 g of beans (Figure 6A) ranged from 109.23 mg to 124.00 mg. In contrast, the magnesium content in 100 g of bean (Figure 6B) ranged from 129.91 g to 137.01 mg, phosphorus content (Figure 6C) from 366.99 mg to 387.00 mg, and the potassium content (Figure 6D) from 1341.20 mg to 1394.06 mg. Statistical analyses revealed statistically significant differences except for potassium, where no differences were found for the two highest doses (Figure 6D, $p = 0.303$). In addition, no differences between the years under analysis were found in terms of the average phosphorus and potassium contents.

**Table 9.** Changes in macroelement content in individual experimental variants of the trial with beans on Haplic LUVISOLS soil type.

| Variant | Carbohydrates in 100 g of Bean (g) | | | | Calcium in 100 g of Bean (mg) | | | | Magnesium in 100 g of Bean (mg) | | | | Phosphorous in 100 g of Bean (mg) | | | | Potassium in 100 g of Bean (mg) | | | |
|---|---|---|---|---|---|---|---|---|---|---|---|---|---|---|---|---|---|---|---|---|
| | 2020 | 2021 | 2022 | $\bar{x}$ | 2020 | 2021 | 2022 | $\bar{x}$ | 2020 | 2021 | 2022 | $\bar{x}$ | 2020 | 2021 | 2022 | $\bar{x}$ | 2020 | 2021 | 2022 | $\bar{x}$ |
| I | 47.00 | 47.00 | 47.00 | 47.00 | 103.00 | 103.12 | 103.59 | 103.24 | 119.99 | 119.96 | 120.00 | 119.98 | 361.15 | 361.23 | 361.09 | 361.16 | 1325.06 | 1325.16 | 1325.02 | 1325.08 |
| II | 49.77 | 49.76 | 49.82 | 49.78 | 109.22 | 109.23 | 109.25 | 109.23 | 129.90 | 129.88 | 129.94 | 129.91 | 367.01 | 367.00 | 366.97 | 366.99 | 1341.20 | 1341.18 | 1341.21 | 1341.20 |
| III | 51.35 | 51.37 | 51.40 | 51.37 | 119.56 | 119.56 | 119.59 | 119.57 | 131.98 | 132.00 | 132.13 | 132.04 | 372.89 | 372.88 | 372.90 | 372.89 | 1393.99 | 1394.00 | 1394.04 | 1394.01 |
| IV | 53.99 | 54.00 | 54.03 | 54.01 | 124.00 | 123.99 | 124.02 | 124.00 | 136.98 | 137.00 | 137.05 | 137.01 | 386.97 | 387.00 | 387.04 | 387.00 | 1394.04 | 1394.05 | 1394.09 | 1394.06 |

$\bar{x}$—average.

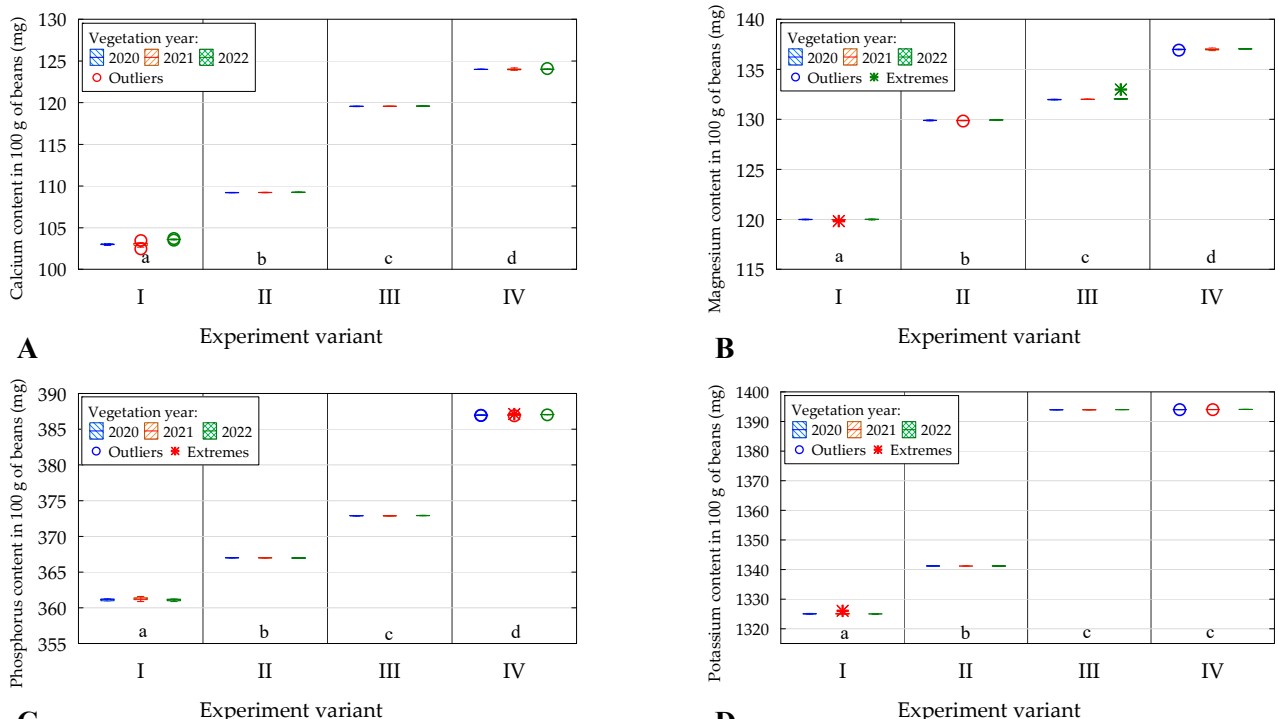

**Figure 6.** Boxplots for microelement contents of the trial with beans on Haplic LUVISOLS soil type: (**A**) calcium, (**B**) magnesium, (**C**) phosphorus, (**D**) potassium. [abcd]—mean values in the groups not containing the same letter differ significantly at $\alpha = 0.05$.

## 4. Discussion

This study was carried out to determine the effect of digestate dose on the yield and quality of multiflora bean seeds.

The bean is a thermophilic plant with a minimum growth temperature of 10 °C, and optimal growth temperature 23 °C as the. During flowering, both too-low (below 15 °C) and too-high temperatures (above 30 °C) hamper the flowering process, leading to flower drop and poor pod setting. Too high a temperature (above 30 °C) during flowering and pod setting can reduce pod length, seed count per pod, and seed weight, accelerate the ripening process, and shorten the growing season. The most suitable cultivation regions are those where the isotherm is 16 °C in June and 18 °C in July. In their study of yields of the Mela and Prosna bean varieties, Dostatny et al. [21] indicated that the 2019 growing season was characterised by an unfavourable pattern of weather conditions, particularly as concerns the prevailing temperatures and the quantity and distribution of precipitation. The average temperature for Radzikowo (52°13′38″ N 20°36′55″ E) was 1.7 °C higher compared to the multi-year average, and was 5.4 °C higher for June, coupled with insufficient precipitation, which resulted in lower yields of the assessed bean varieties (Mela—2.26 kg ha$^{-1}$, Prosna—2.74 kg ha$^{-1}$). The observations made in their study as well as in those by other authors allow us to conclude that the meteorological conditions during our study were suitable for the sowing and harvesting of bean seeds.

In the conducted study, the thousand-seed weight and nutrient content (e.g., the content of protein, fats, carbohydrates, calcium, magnesium, phosphorus, and potassium) in 100 g of seeds were treated as determinants of seed quality. Dostatny et al. [21] and Wondołowska-Grabowska [22] also considered the thousand-seed weight and nutrient content in 100 g of seeds to be the underlying parameters of bean seed quality.

At a row spacing of 40 cm, the average pod count per plant for the three growing periods ranged from 8.33 ÷ 11.33 depending on the applied digestate dose. A similar pod count per plant was recorded by Dostatny et al. [21]. In that study [21], the count reached 11; Merga [23], in contrast, obtained a higher pod count per plant (13.07 ÷ 21.28), also at a

row spacing of 40 cm. A generally higher pod count was recorded when the beans were sown at a row spacing of 60 cm, while a lower count was obtained at a row spacing of 30 cm. A wider row spacing and less competition between the plants are both factors likely to influence the pod count per plant [23]. Meanwhile, the seed count per plant ranged from 31.00 to 43.00, depending on the digestate dose. In the study by Dostatny et al. [21], the seed count per plant was 41. Merga [23], in turn, analysed the seed count per pod, obtaining results ranging from 3.74 pcs to 4.56 pcs depending on the plant variety. Competition for sunlight and nutrients due to insufficient row spacing may result in a lower seed count per pod [23,24]. Higher values of pod count per plant were obtained by Peeters et al. [25], while Sosnowski [26] claimed that the seed count per pod was a variable parameter depending on the genetic characteristics of both the species and varieties as well as on the location of the pod on the stem tiers. According to these authors, the number of pods per plant and number of seeds per plant obtained in our research are suitable for the sown bean variety. In addition, fertilization with digestate, especially the highest dose, had an impact on the obtained values, indicating the promise of using digestate as a fertilizer.

In his study, Merga [23] found that the row spacing and plant variety, along with the interaction of both these parameters, influenced the yield and macroelement content of the common bean. He recorded a yield of 2.24 t ha$^{-1}$ (Nasir variety) and 2.01 t ha$^{-1}$ (Goberesha variety) with beans sown at a row spacing of 40 cm. For the local variety, the yield was 1.6 t ha$^{-1}$ at a row spacing of 50 cm. In contrast, Kotecki et al. [27] obtained a yield ranging from 1.85 t ha$^{-1}$ to 2.69 t ha$^{-1}$ depending on the common bean variety.

The yield recorded in this study was comparable, averaging from 2.07 t ha$^{-1}$ to 2.61 t ha$^{-1}$ depending on the applied digestate dose. Similar yield figures for beans were reported by Kapusta [28]. The obtained yield and thousand-seed weight are appropriate for this bean variety, and depend on the applied dose of digestate. The obtained results show the fertilizer potential of the digestate.

The thousand-seed weight reported in this study, depending on the applied digestate dose, reached 171.39 g $\div$ 184.44 g in average terms. Similar results were obtained by Dostatny et al. [21] (175 g), while the results recorded by Dahmardeh et al. [29], Masa et al. [30], and Merga [23] were higher, reaching around 450 g depending on the studied variety. Sinkovič et al. [31] obtained a higher thousand-seed weight in their studies, which ranged from 411 g to 552 g depending on the plant variety. Thousand-seed weight may be influenced by the differences among bean genotypes. In addition, this parameter has been found to increase with wider row spacings. More specifically, the supply of assimilates retained in the seeds was found to improve in widely spaced plants [29,30]. Sosonowski [26] reported that thousand-seed weight was related to the actual seed size, and ranged between 150 g and 700 g for the existing Polish varieties.

This study found that the digestate dose used for fertilisation purposes had an effect on the nutrient content in 100 g of bean seeds. Montemurro et al. [32] and Pan et al. [33] concluded that by applying the digestate, macroelements were supplied to the soil, in particular nitrogen, phosphorus, and potassium. The results of a study by Panuccio et al. [34] suggest the use of digestate as an alternative to of mineral fertilisers. Kapusta [28] stated that protein was an essential component of bean seeds, with its content reaching 23%. In addition, Sosnowski [26] found that bean protein is characterised by a high biological value, and that the seeds contain relatively high amounts of phosphorus and iron. In contrast, the seed fat content was found to reach low levels of 1.8–2.6%.

In a study by Prusiński et al. [35], a protein content of 19.7–21.2% was obtained in runner beans, which depended on the variety. In turn, Wawryka et al. [36] reported a protein content of 17% in the analysed beans.

In our study, the average protein content ranged from 19.83% to 21.45% depending on the applied digestate dose, while the average fat content varied from 0.47% to 0.61%.

In their study, Wieczorek et al. [37] found that the highest carbohydrate content in white bean seeds reached 42.56 mg g$^{-1}$ dry weight. In contrast, Dostatny et al. [21] reported the carbohydrate content at 60 g in 100 g of bean seeds, while Sahasakul et al. [38] found

that the carbohydrate content in bean seeds ranged from 33.12 g 100 g$^{-1}$ dry weight to 77.39 g 100 g$^{-1}$ dry weight depending on the variety. Hayat et al. [39] concluded that the carbohydrate content of bean seeds depended on the variety and was within the range of 50.4% ÷ 68.09%. In contrast, lower carbohydrate values (16.18–40.7%) were obtained by Brigide et al. [40]. In the study presented in this paper, the carbohydrate content in bean seeds, depending on the experimental variant, ranged from 49.78 g to 54.01 g per 100 g of bean seeds for the three growing seasons. A slightly higher carbohydrate content (53.3 g 100 g$^{-1}$ dry weight ÷ 67.1 g 100 g$^{-1}$ dry weight) was obtained by Shahrajabian et al. [41].

In their study, Kotecki et al. [27] determined the calcium content in common bean seeds to be about 100 g. They obtained the following values in the bean varieties under analysis: 0.15% to 0.22% for calcium, 0.20% ÷ 0.24% for magnesium, 0.42% ÷ 0.45% for phosphorus, and from 1.48% to 1.55% for potassium. To compare, Prusiński et al. [35], in their study of runner bean varieties, reported a calcium content of 0.635% ÷ 0.687%, magnesium content ranging from 0.277% to 0.291%, phosphorus content of 0.513% ÷ 0.552%, and potassium content ranging from 2.614% to 2.928%. Wawryka et al. [36] reported a calcium content in 100 g of seeds of 4%, magnesium content of 10%, phosphorus content of 14%, and potassium content of 15%, all for the red bean variety. Herrera-Hernádez et al. [42] obtained the following macroelement contents: 0.14% to 0.29% for calcium, 0.13% to 0.40% for phosphorus, and 0.41% to 1.32% for potassium, depending on the bean variety under analysis. In contrast, García-Caparrós et al. [43] obtained a phosphorus content from 295.59 mg 100 g$^{-1}$ dry weight to 1623.41 mg 100 g$^{-1}$ dry weight depending on the bean variety.

In our study, the average calcium content in 100 g of beans was determined as 109.23 mg to 124.00 mg and the average magnesium content as 129.91 mg to 137.01 mg. In contrast, the average phosphorus content in 100 g of beans ranged from 366.99 mg to 387.00 mg, and the average potassium content in 100 g of beans reached 1341.20 mg ÷ 1394.06 mg. Comparable values in 100 g of bean were obtained by Dostatny et al. [21]: calcium—143 mg, magnesium—140 mg, phosphorus—407 mg, and potassium—1406 mg. Similar values were obtained by Sahasakul et al. [38] for calcium (86.40 mg 100 g$^{-1}$ to 242.17 g 100 g$^{-1}$ dry weight), magnesium (115.46 g 100 g$^{-1}$ to 224.81 g 100 g$^{-1}$ dry weight), and potassium (915.34 g 100 g$^{-1}$ to 1517.36 g 100 g$^{-1}$ dry weight).

Based on the findings of previous the authors, the nutritional content of the bean seeds in this study is adequate for the studied variety. The obtained results confirm the possibility of using digestate as a fertilizer instead of mineral fertilizers.

## 5. Conclusions

Based on the conducted study, the following conclusions can be drawn:

1. The digestate dose influences the yield and thousand-seed weight. Among the three digestate doses analysed, the highest is recommended. The highest tested dose of digestate resulted in a high seed yield and high macroelement content in the seeds.
2. The digestate dose is a factor in the protein content of multiflora bean seeds.
3. The digestate dose influences the macroelement content. Statistically significant differences were found between the different doses of digestate.
4. The highest dose contains the most nitrogen, and as such is the recommended dose. It affects the physical parameters of the plants as well as the protein, fat, and macroelement contents of the beans.
5. The above experiment was established on the Haplic LUVISOLS soil type according to the FAO classification in Poland. Completely different results could be obtained on other soil types; thus, it is necessary to carry out further tests on other soil types.

**Author Contributions:** Conceptualization, M.K. and S.P.; methodology, S.K.; software, A.B.; validation, M.K. and S.P.; formal analysis, E.L.; investigation, M.K.; resources, A.P.; data curation, P.F.; writing—original draft preparation, M.K.; writing—review and editing, S.P.; visualization, A.Z.A.; supervision, S.K.; project administration, E.L.; funding acquisition, A.P. All authors have read and agreed to the published version of the manuscript.

**Funding:** This research received no external funding.

**Institutional Review Board Statement:** Not applicable.

**Data Availability Statement:** All data are contained within the article.

**Conflicts of Interest:** The authors declare no conflicts of interest.

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
