# Peer review of "Analysis of the Physico-Chemical Properties of Bean Seeds after Three Years of Digestate Use"

_agriculture, doi:10.3390/agriculture14030486_

Round 1

Reviewer 1 Report (Previous Reviewer 4)

Comments and Suggestions for Authors

The authors made the requested changes and additions to the manuscript.

The manuscript meets high standards for publication in the journal "Agriculture"

Author Response

Thank you for your favorable consideration of the manuscript for publication. 

Reviewer 2 Report (Previous Reviewer 3)

Comments and Suggestions for Authors

Manuscript agriculture-2872308 - The analysis of physico-chemical properties of bean seeds after three years of digestate use 

The manuscript is about the effects of digestate fertilization on the yield and properties of been seeds.

The manuscript has been improved in all the items. However, the manuscript still needs more improvements, as follows.

Abstract 

The abstract should refer clearly the experiment done and its results. The authors should include information about the experimental field design. The results in the abstract are listed as average instead of being to the effect of each treatment.

In line 29 what is the meaning of “analysis”? 

The last phrase should be rewritten, the authors should clarify what are the source of the benefits of this dose of digestate in increasing the yield. Also, in the statement should be included the statistical difference of this treatment in comparison with the other treatments. 

Introduction

Lines 39-41 – The authors have to explain why the digestate is important to replace for example N fertilizers. The phrase “liquid fertilizers….by plants” is decontextualized.

Line 44- The authors should explain better what is the negative impact of using mineral fertilizers, e.g. what are the negative impacts that the digestate will overcome. 

Lines 106-107 – the hypothesis should be clear. What is the meaning of the “size of the digestate?

Lines 108-112- This paragraph is not part of the hypothesis. It should be placed before. 

Overall, all the paragraph (lines 105-117) should be rewritten. 

Material and methods

The amount of N, P and K applied by the digestate in each treatment should be referred.

The experimental design of the field experiment should be explained. How many replicates for each treatment? The size of each plot? 

The methods used in the soil analysis are not described. 

The amount of N in the digestate is not displayed in table 1 , why?

Tables 2 and 3- The phosphorus and potassium are expressed in P, K? or in P2O5, K2O.

Why use mg/100g instead of mg/kg? 

The authors should refer the fertility class for each element of the soil used in the experiment (Table 2). 

Table 3 – The amounts of P, K, and Mg and the pH value are the average values of all the treatments? 

Why, the initial P, K and Mg values (Table 2) are so high compared with those in 2020, 2021 and 2022 (table 3)? The digestate and the mineral fertilization done seems to apply to soil enough P to prevent this decrease. The decrease in the soil P content observed is the same in all the treatments? 

Why did the plants have higher yield in the treatment IV? Can be, because they have more nutrients, such as N to their needs? It’s important to present the amount of N applied by the digestate. 

Discussion

This section should be improved because the authors only listed the results obtained and their comparison with other works, which should be included in the Results section. In this section the authors have to present their opinion about the results of this work contextualized with the opinion of other authors about the same or similar topics.  

Conclusions

Lines 803-805 should be removed.

After the improvements in the manuscript, in particular in the discussion section, the conclusions have to be rewritten. 

Comments on the Quality of English Language

editing should be done by a native English speaking 

Author Response

Thank you for your critical review. It will certainly contribute to the scientific quality of the submitted manuscript. The following corrections have been made to the manuscript:  

The abstract has been completely modified.  

It was explained why digestate is important to replace mineral fertilisers. The expression 'liquid fertilisers ... by plants" has been removed. 
The negative effects of mineral fertilisers have been clarified. 
The word "size" has been replaced by "volume". 
Lines 108-112 - previously rioted.  

The design of the field experiment is described - number of replicates, size of each plot. 
Methods of soil analysis are included. 
The amount of nitrogen in Table 1 has been completed. 
In Tables 2 and 3, phosphorus and potassium are expressed as P205 and K2O. 
The units used in the tables are based on reports received from an accredited laboratory. 
Table 3 - these are average values. 
The decrease in macronutrients is related to uptake by the plants. 
Higher yields in variant IV of the experiment are associated with higher nitrogen and macronutrient content. 

Round 2

Reviewer 2 Report (Previous Reviewer 3)

Comments and Suggestions for Authors

Manuscript agriculture-2872308 - The analysis of physico-chemical properties of bean seeds after three years of digestate use 

The manuscript is about the effects of digestate fertilization on the yield and properties of been seeds.

The manuscript has been improved. However, the improvements needed in the section Discussion has not been done as indicated in the last revision:

“Discussion

This section should be improved because the authors only listed the results obtained and their comparison with other works, which should be included in the Results section. In this section the authors have to present their opinion about the results of this work contextualized with the opinion of other authors about the same or similar topics. “ 

So, in my opinion for the publication of the manuscript the section Discussion should be rewritten since the authors did not revise any part of this section.

Author Response

Thank you for your review. 
The Discussion chapter has been rewritten. We have added our opinion on the research conducted. 

This manuscript is a resubmission of an earlier submission. The following is a list of the peer review reports and author responses from that submission.

Round 1

Reviewer 1 Report

Comments and Suggestions for Authors

The article deals with significant and interesting issues about beans yield depending on the digestate dose applied to different doses. The work is mainly related to issues of Agronomic science. Study confirms that the best average results were obtained for a dose of 50.000 l.ha-1. The results of the work may have some practical significance in improving soil fertilization in agriculture.

I would like to make a few comments about the text of the article:

1. In my opinion, it would be useful to present the design of the field experiment in the Materials and Methods section. This can be done using a graphical expression. This would help to understand the whole experiment, number of samples, number of replicates etc.

2.      It would be useful to provide more information about the physical properties of the digestate raw materials in Materials and Methods section. It was mentioned from which materials (in percentage terms) the digestate used for the production of biogas was obtained (Line 128. 129), but there is not enough information about the physical form (amount of dry materials etc.) before it was incorporated during fertilization. It is not clear whether any additional steps are required before spraying.

3.      Descriptions or references to standards of chemical composition evaluation could be provided in Materials and Methods chapter. One gets the impression that all methods are created by the author.  

4.      Explanation is submitted as to why the results of the heavy metal Cadmium, Nickel, Chromium  tests results were the same in Table 2. But, I would still suggest specifying the heavy metal norms for organic fertilizers.  Can be used Regulations (EU) 2019/1009 of the European Parliament and of the councils laying out rules on fertilizer products (EC) No 1069/2009 and (EC) No 1107/2009 (Available online: https://eur-lex.europa.eu/legal-content/EN/TXT/?uri=celex%3A32019R1009). The comparison could allow us to say that the digestate can be used as a fertilizer.

5. The conclusions should be more concrete and specific comparable quantitative results could be use. There is a grammatical error (unclear dot meaning) in the conclusions (Line 525). In my opinion it is worth mentioning other variants, or which of them are not recommended.

Author Response

  1. In my opinion, it would be useful to present the design of the field experiment in the Materials and Methods section. This can be done using a graphical expression. This would help to under-stand the whole experiment, number of samples, number of replicates etc.

Answer:

We agree with the reviewer that a graphical expression would have made the manuscript even more readable. We have included all the information needed for the experiment in the Materials and Methods chapter, which is why we did not do a graphical expression.

  1. It would be useful to provide more information about the physical properties of the digestate raw materials in Materials and Methods section. It was mentioned from which materials (in per-centage terms) the digestate used for the production of biogas was obtained (Line 128. 129), but there is not enough information about the physical form (amount of dry materials etc.) before it was incorporated during fertilization. It is not clear whether any additional steps are required be-fore spraying.

Answer:

We added the sentence: “The digestate was applied in liquid form using spill hoses”.

No additional treatments are needed before spraying.

  1. Descriptions or references to standards of chemical composition evaluation could be provided in Materials and Methods chapter. One gets the impression that all methods are created by the au-thor.

Answer:

In the Materials and Methods chapter, we have included the standards according to which the re-search methodology was developed.

  1. Explanation is submitted as to why the results of the heavy metal Cadmium, Nickel, Chro-mium tests results were the same in Table 2. But, I would still suggest specifying the heavy metal norms for organic fertilizers. Can be used Regulations (EU) 2019/1009 of the European Parlia-ment and of the councils laying out rules on fertilizer products (EC) No 1069/2009 and (EC) No 1107/2009 (Available online: https://eur-lex.europa.eu/legal-content/EN/TXT/?uri=celex%3A32019R1009). The comparison could allow us to say that the di-gestate can be used as a fertilizer.

Answer:

We have circled the heavy metal content standards for organic fertilisers below the table.

  1. The conclusions should be more concrete and specific comparable quantitative results could be use. There is a grammatical error (unclear dot meaning) in the conclusions (Line 525). In my opin-ion it is worth mentioning other variants, or which of them are not recommended.

Answer:

The full stop has been removed. We have corrected the conclusions.

Reviewer 2 Report

Comments and Suggestions for Authors

The study which aims to analyze the physicochemical properties of bean seeds after three years of digestate use is a very interesting study which is very well structured and keeps a connection between the objective, the methodology, the results and the discussion. 

I consider that the study should have a hypothesis, since it is not currently found in the document.

In general, I consider that it should pay a lot of attention to the way units are cited and the use of unnecessary decimals. For example, in line 24 there is a number 2.61, I consider that it is not necessary to use two decimals, likewise, the unit t*ha-1 should go in t ha-1, eliminating the symbol that accompanies the unit. 

L25: The unit would be L ha-1

I think you should include more keywords, for example the scientific name of the bean species you are using. 

You should look for a better way to show the information in table 1, I suggest a figure. In table 2 you should place the standard errors of each of the means. In table 3 you should place a column where you put the units of each of the variables. 

A in figure 1 should be eliminated. The meaning of the letters a, b, c, d and I, II, III and IV should be explained in the legend. 

In Table 6, the means and standard errors should be placed together with the letters that identify the differences between the means. 

In figure 2 it should be adjusted and explain why it is identified with C and D. 

Author Response

  1. I consider that the study should have a hypothesis, since it is not currently found in the document.

Answer:

We have added a hypothesis at the end of the Introduction section.

  1. In general, I consider that it should pay a lot of attention to the way units are cited and the use of unnecessary decimals. For example, in line 24 there is a number 2.61, I consider that it is not nec-essary to use two decimals, likewise, the unit t*ha-1 should go in t ha-1, eliminating the symbol that accompanies the unit.

Answer:

We have corrected the units as suggested by the Reviewer.

We have left one decimal place where possible. However, in some data we had to leave two decimal places in order to make the changes visible.

  1. L25: The unit would be L ha-1

Answer:

We have improved the unit.

  1. I think you should include more keywords, for example the scientific name of the bean species you are using.

Answer:

We have added the following keywords: bean Phaseolus coccineus seeds, protein content, microele-ments, macroelements.

  1. You should look for a better way to show the information in table 1, I suggest a figure. In table 2 you should place the standard errors of each of the means. In table 3 you should place a column where you put the units of each of the variables.

Answer:

Instead of Table 1, we have included two figures showing the distribution of temperature and precip-itation. In Table 1 we added a column with stadard deviation and in Table 4 we added a column with units.

  1. A in figure 1 should be eliminated. The meaning of the letters a, b, c, d and I, II, III and IV should be explained in the legend.

Answer:

We have corrected the numbering of the drawings. I, II, III, IV are on the X axis, which is signed as Ex-periment variant. The letters a, b, c, d are explained below the figure.

  1. In Table 6, the means and standard errors should be placed together with the letters that identify the differences between the means.

Answer:

We have not placed the averages and standard errors together with the letters that identify differ-ences between the averages, as the statistical analysis is presented in Figure 5.

  1. In figure 2 it should be adjusted and explain why it is identified with C and D.

Answer:

We have renumbered the drawings. Each drawing has a different number.

Reviewer 3 Report

Comments and Suggestions for Authors

The manuscript is about the effects of digestate fertilization on the yield and properties of been seeds.

The manuscript needs to be improved in all the items. Honestly, I cannot do a proper revision of the manuscript because all the sections are very confused. The authors made statements without a previous explanation and contextualization. The hypothesis of the work is not referred.   The objectives are not clear. The experimental design was not clearly explained. The parameters of the anaerobic digestion were not mentioned e.g. temperature, retention time. The type of digestate used: the liquid fraction? The moisture of the digestate? The N content of the digestate is not referred. 

The results obtained by the authors are not clearly explained and discussed. The authors simply compared their results with those in the references. 

So, in my opinion the publication of the manuscript needs major revisions in all the sections.

Overall, it is not possible to evaluate the interest of this work for readers. 

Author Response

Answer:

The purpose of the conducted research is given and supplemented with a hypothesis at the end of Chapter 1. Information on the needs of liquid fertilizer application in bean cultivation was completed. The parameters of anaerobic digestion were not studied in this manuscript. Information on the form of the digestate used was also completed. The Material and Methods chapter was written according to the rules of writing chapters of this type. The posted description of the research allows to post-pone this research. The international standards on the basis of which the measurements were made have also been included. Data on the pH and macronutrient content of the soil before sowing and after harvesting the beans have been completed.

Reviewer 4 Report

Comments and Suggestions for Authors

The topic addressed by the authors of the manuscript is interesting for the wider scientific community. The topic is challenging and demanding. Unfortunately, the authors did not respond to this challenge.

Significant improvements need to be made in the manuscript. The following shortcomings were observed, which need to be corrected, in order for this manuscript to meet the high standards of publishing papers in the journal.

Title: satisfies.

Abstract: It is necessary to state: (1) on which type of soil the research was carried out; and (2) how many treatments there were.

Introduction: It is necessary to emphasize the problems due to which the application of digestates, as fertilization agents, is limited.

The last sentence in the introduction is not appropriate. It is necessary to state: why this research was carried out on beans and what is the significance (application) of this research.

Materials and Methods: Оn which type of soil the research was carried out? The research results are related to: (1) the plat species; (2) treatments; and (3) SOIL TYPE! State the type of soil used in the experiments according to the FAO soil classification.

Are there any photos (pictures) of the show? If they exist, why are they not in the manuscript?

Results: The data source for the meteorological data is not listed in the Material and Methods. Then, it is customary to present the meteorological data in Table 1 graphically (first of all, it is more transparent and takes up less space in the work).

Line 215-216: [Soil analysis for acidity and macroelement content was carried out before sowing and after harvesting the multiflora bean seeds.] This sentence has a place in materials and methods.

Line 217-233: This data should be presented in a table.

The titles of tables 3, 4, 5, 6, 7 and 8 are not sufficiently explanatory. It should read, for example: Results of GLM analysis of a trial with beans on soil type ..... The same goes for figures.

Discussion: It should be emphasized that these results justify the application of digestate for bean fertilization on a certain type of soil, in Poland! Confirmation of these results on other types of soil, with other cultivated crops and in other environments (spaces) should be proven (confirmed) in other research.

References: Of the 37 references cited in the manuscript, as many as 4 are from 2021 and 2022.

Author Response

  1. Abstract: It is necessary to state: (1) on which type of soil the research was carried out; and (2) how many treatments there were.

Answer:

We specified the soil type according to the FAO classification and the number of repetitions.

  1. Introduction: It is necessary to emphasize the problems due to which the application of diges-tates, as fertilization agents, is limited.

Answer:

We have given the reasons for the bean study at the end of Chapter 1.

  1. The last sentence in the introduction is not appropriate. It is necessary to state: why this research was carried out on beans and what is the significance (application) of this research.

Answer:

At the end of Chapter 1, we specified why the research was conducted on beans and its significance.

  1. Materials and Methods: Оn which type of soil the research was carried out? The research results are related to: (1) the plat species; (2) treatments; and (3) SOIL TYPE! State the type of soil used in the experiments according to the FAO soil classification.

Answer:

We specified the type of soil according to the FAO classification on which the experiment was set up.

  1. Are there any photos (pictures) of the show? If they exist, why are they not in the manuscript?

Answer:

Thank you for your valuable consideration. Unfortunately, we do not have photos of this experience. We will include them in future manuscripts.

  1. Results: The data source for the meteorological data is not listed in the Material and Methods. Then, it is customary to present the meteorological data in Table 1 graphically (first of all, it is more transparent and takes up less space in the work).

Answer:

In Materials and Methods, we listed the source of the meteorological data. We have transformed Ta-ble 1 into a figure.

  1. Line 215-216: [Soil analysis for acidity and macroelement content was carried out before sowing and after harvesting the multiflora bean seeds.] This sentence has a place in materials and met-hods.

Answer:

We have removed this sentence.

  1. Line 217-233: This data should be presented in a table.

Answer:

The data provided in line 217-233 only apply to the description of Variant I (control facility). In addi-tion, we have included tables with measurements of soil pH and macronutrient content in the soil before sowing and after harvesting.

  1. The titles of tables 3, 4, 5, 6, 7 and 8 are not sufficiently explanatory. It should read, for example: Results of GLM analysis of a trial with beans on soil type ..... The same goes for figures.

Answer:

Captions over the table and under the figures have been corrected as suggested by the Reviewer.

  1. Discussion: It should be emphasized that these results justify the application of digestate for bean fertilization on a certain type of soil, in Poland! Confirmation of these results on other types of soil, with other cultivated crops and in other environments (spaces) should be proven (con-firmed) in other research.

Answer:

We have swept into the conclusions a point about testing on a specific soil type.

We confirm that this manuscript has not been published elsewhere and is not under consideration by another journal.

We have approved the manuscript and agree with its submission to Agriculture” – special issue ”Ef-ficient Use of Irrigation and Fertilizer to Increase Crop Yield.